# The Functional Identification of the *CYP2E1* Gene in the Kidney of *Lepus yarkandensis*

**DOI:** 10.3390/ijms26020453

**Published:** 2025-01-08

**Authors:** Dingwei Shao, Ke Sheng, Bing Chao, Yumei Tong, Renjun Jiang, Jianping Zhang

**Affiliations:** Xinjiang Production and Construction Corps Key Laboratory of Protection and Utilization of Biological Resources in Tarim Basin, College of Life Science, Tarim University, Alar 843300, China; weishao2021@outlook.com (D.S.); 15954101003@163.com (K.S.); chaobing5824@163.com (B.C.); tongym071459@163.com (Y.T.); 120050057@taru.edu.cn (R.J.)

**Keywords:** *Lepus yarkandensis*, *CYP2E1*, stress resistance, yeast expression, oxidative stress lethal threshold

## Abstract

This study aims to identify the function of the *cytochrome P450 2E1* (*CYP2E1*) gene in the kidneys of *Lepus yarkandensis*. CYP2E1 is a significant metabolic enzyme involved in the metabolism of various endogenous and exogenous compounds and is associated with the occurrence and progression of multiple diseases. Given *L. yarkandensis*’s ability to survive in the extremely arid *L. yarkandensis*, we hypothesize that CYP2E1 in its kidneys plays a crucial role in adaptability. Through molecular cloning and sequence analysis, we discovered that the *CYP2E1* gene of *Lepus yarkandensis* encodes a protein of 493 amino acids. The 493-amino acid protein encoded by the *Lepus yarkandensis CYP2E1* gene shows 13 amino acid variation sites compared to the homologous protein in *Oryctolagus cuniculus*. The protein is primarily localized to the endoplasmic reticulum membrane and lacks transmembrane structures. In the yeast expression system, the heterologous expression of the *CYP2E1* gene enhanced the yeast’s tolerance to drought, salinity, and high temperatures, achieved by increasing antioxidant enzyme activity and reducing levels of oxidative stress markers. Additionally, this study identified a “Yeast Oxidative Stress Lethal Threshold (Yeast OSLT)” under specific stress conditions. Once this threshold is exceeded, the cell’s antioxidant defense system can no longer maintain cellular homeostasis, leading to massive cell death. Although CYP2E1 did not change this threshold, it contributed to cell survival to some extent. These findings not only reveal the function of *L. yarkandensis* CYP2E1 in stress adaptation but also provide valuable molecular insights into its survival strategy in extreme environments.

## 1. Introduction

Cytochrome P450 (CYP450) is a class of membrane-bound heme proteins found in plants, animals, microorganisms, and humans, involved in the metabolism of drugs, steroids, fat-soluble vitamins, and many other types of chemicals [1]. CYP2E1, a member of the P450 family, is highly expressed in the liver and is also expressed to a certain extent in the kidneys [2]. Under normal expression levels, CYP2E1 has multiple positive effects on cells, including the metabolism of potentially toxic substances such as pro-carcinogens, pro-drugs, and certain medications, aiding the body in clearing these substances [3]. It also participates in the regulation of diseases such as diabetes, fatty liver, and cancer, demonstrating its broad physiological activity. Furthermore, CYP2E1 interacts with the nuclear receptor PPARα to co-regulate obesity, and *CYP2E1* gene knockout mice show significantly improved obesity and dyslipidemia under high-fat diets. CYP2E1 also participates in increasing oxidative metabolic rates, promoting the browning of white adipose tissue and thermogenic gene expression, enhancing energy expenditure [4]. As a drug-metabolizing enzyme, CYP2E1 is responsible for metabolizing various endogenous and exogenous small molecules [5]. It is also involved in detoxification processes, converting toxic chemicals into non-toxic or easily excretable forms [6]. The expression of CYP2E1 is also related to the regulation of oxidative stress responses, and its genetic polymorphisms enable cells to adapt to different physiological and pathological states. CYP2E1 can indirectly maintain redox balance by activating antioxidant pathways such as Nrf2, better equipping cells to cope with various adverse environments [7]. *L. yarkandensis*, as a species adapted to extreme environments, survives and reproduces in the harsh conditions of arid, hot, and saline, indicating it may possess unique physiological and molecular adaptive mechanisms. The kidney, as an essential excretory organ, plays a vital role in maintaining homeostasis. Therefore, studying the expression patterns and functions of *CYP2E1* in the kidneys of *L. yarkandensis* is significant for understanding their adaptability in extreme environments.

*L. yarkandensis* is a rare and endemic species in China, mainly distributed in the oasis and desert areas in Xinjiang. As a species living in extremely arid environments, *L. yarkandensis* has shown unique adaptive characteristics. Although the genomic information of *L. yarkandensis* is currently relatively limited, research on its genetic variation, population structure, and phylogenetic relationships has gradually progressed. For example, using specific length amplified fragment sequencing (SLAF-seq) technology, researchers have assessed the genetic diversity of 76 *L. yarkandensis* and explored population differentiation and evolutionary processes based on single-nucleotide polymorphism (SNP) markers [8]. The complete mitochondrial genome sequence analysis of *L. yarkandensis* reveals its uniqueness in phylogenetic evolution, which may be related to its ability to adapt to extreme environments [9]. In addition, research on the regulation of gene expression in *L. yarkandensis* is ongoing, especially in terms of the expression regulation of water reabsorption genes and transcription factors in the kidneys. Studies have shown that the expression patterns of aquaporins (AQPs) in the kidneys of *L. yarkandensis* are very closely related to the secretion, absorption of water, and the balance of water inside and outside of cells [10]. Transcription factors such as *HIF1A*, *NFATc1*, and *NF-κB1* have also been focused on. The expression of these transcription factors in the kidneys of *L. yarkandensis* and their functions in adapting to extremely arid environments have been studied [11]. The *TLR4* gene, which is closely related to intestinal immunity and inflammatory responses, may have a unique role in the regulation of immunity and inflammation in *L. yarkandensis* [12]. The study of these genes helps to reveal the molecular mechanisms by which *L. yarkandensis* adapts to extremely arid environments. In summary, the research on the adaptive evolution of *L. yarkandensis* not only reveals its adaptive mechanisms at the genetic level but also provides important molecular insights into the survival strategies of the species in extreme environments. These studies have significant scientific value and practical significance for the conservation of biodiversity and the maintenance of ecosystem stability.

In terms of cloning stress-resistant genes in animals, researchers have made significant progress in various animals. For example, by overexpressing the human *CYP2E1* gene, the metabolism and removal rate of volatile environmental pollutants such as trichloroethylene, chloroform, and benzene have been significantly improved, thereby enhancing the potential of phytoremediation technology in dealing with volatile organic pollutants [13]. Camels have been studied for their ability to survive under extreme arid conditions, and scientists have cloned multiple genes related to drought resistance [14]. Similarly, the cold tolerance of *red-claw crayfish* has attracted the attention of many researchers, and several genes related to low-temperature adaptability have been discovered through gene cloning techniques [15]. In addition, yaks, as unique animals in high-altitude areas, not only have unique characteristics in adapting to high-altitude environments but have also become research objects due to their potential anti-tumor capabilities [16]. New methods for cloning resistance genes from wild relatives of crops by sequence capture and association genetics provide an economical and environmentally sustainable method for crop disease protection [17].

Yeast expression vectors, as an efficient eukaryotic expression model, play an important role in the field of biotechnology, especially in plant gene function research, where they have been widely verified and applied [18]. Compared with knockout and overexpression methods commonly used in animal gene function research, the yeast expression system provides a simpler, faster, and lower-cost method for gene function verification. Yeast, especially *Saccharomyces cerevisiae*, is the first eukaryote to have its genome sequenced, with a small genome and relatively clear gene functions, and its genetic background has been well understood after years of research, enabling scientists to use yeast to study the gene functions of higher animals and plants [19]. In plant gene function research, the yeast heterologous functional complementation method plays an important role, verifying the function of heterologous genes through yeast mutants or cloning heterologous genes through library screening, providing a shortcut for gene cloning and functional verification in higher eukaryotes [20]. In contrast, in animal gene function research, common methods such as gene knockout and overexpression, while direct, have many limitations in practice, such as a high cost, complex operation, and a long cycle. In comparison, the yeast expression system can provide a relatively simple and low-cost platform for gene function verification. Yeast expression vectors such as PYES2 contain an inducible promoter GAL that can induce the expression of foreign proteins in the presence of galactose and repress expression in the presence of glucose, providing researchers with flexible control means [21]. The yeast expression system not only plays an important role in plant gene function research but also shows great potential and advantages in animal gene function research. By using the yeast expression system, researchers can verify gene functions in a relatively simple and controllable environment, providing a new avenue for gene function research. For example, by expressing the salt-tolerant gene of sea buckthorns in Saccharomyces cerevisiae, researchers have successfully revealed the function of the gene under salt stress. Ginsenoside Rg1 can delay the chronological aging process of yeast cells by regulating CDC19 and SDH2, two proteins related to cellular metabolism, thereby providing theoretical support for biological anti-aging [22]. Using the yeast functional screening system to explore multiple salt tolerance-related genes in the halophyte *Atriplex canescens* has also been carried out [23]. The *BpERF* gene plays an important role in the abiotic stress response of *Betula platyphylla* and can improve the tolerance to salt and drought stress in transgenic yeast cells [24]. As a recombinant expression system, yeast has successfully expressed the wheat gene *TaCRK68-A*, thereby revealing its key role in enhancing the plant’s tolerance to heat, drought, cold, and salt stress [25].

This study is dedicated to exploring the function of the *CYP2E1* gene in the kidneys of *Lepus yarkandensis* and attempting to reveal the possible mechanism by which it functions in adaptation to extremely arid environments. Given that the *CYP2E1* gene has been proven to play a crucial role in processes such as metabolism, detoxification, and oxidative stress responses and is expressed in the kidneys of *Lepus yarkandensis*, we reasonably hypothesize that the *CYP2E1* gene may play a significant role in the adaptive evolution of *Lepus yarkandensis*. To test this hypothesis, we plan to utilize techniques such as molecular cloning, sequence analysis, and yeast expression systems to meticulously explore the sequence characteristics, physicochemical properties of the *CYP2E1* gene, and the impact of its heterologous expression in yeast on enhancing stress tolerance. We anticipate that this study can furnish valuable data and information for uncovering the molecular mechanism of *Lepus yarkandensis’s* adaptation to extreme environments, provide certain scientific references for understanding the adaptive strategies of organisms under extreme conditions, and thus contribute to the accumulation of knowledge related to biodiversity conservation and ecosystem stability to a certain extent.

## 2. Results

### 2.1. Sequence Analysis of L. yarkandensis CYP2E1 Gene

The expected band of *L. yarkandensis CYP2E1* was obtained by PCR amplification using cDNA from kidney tissue as a template (Figure 1A). After sequencing at Shanghai Shenggong, the complete nucleotide sequence of *CYP2E1* was 1541 bp (Figure 1A), of which 1482 bp ORFs encoded a protein of 493 amino acids. Compared with the *O. cuniculus*, the total number of amino acids in the CYP2E1 protein did not change, but there were differences in the number of individual amino acids (Figure 1B). There were 39 base mutations in the gene encoding sequence (Appendix A), 13 of which resulted in amino acid changes, which may affect the structure and function of the protein (Figure 1C).

A BLAST search of the CYP2E1 protein sequence on the NCBI protein website showed high similarity with other CYP2E1 proteins. A total of 13 species showed 86.89% identity using DNAMAN 8.0 software (Figure 2A). CYP2E1 proteins and homologs from several other animals were used for phylogenetic analysis (Figure 2B).

### 2.2. Physicochemical Properties and Predicted Structure of CYP2E1 Protein

This study first used the ProtParam online tool to analyze the physicochemical properties of the CYP2E1 protein. The molecular weight of the protein is about 57 KDa, and the theoretical isoelectric point is 8.37. The CYP2E1 protein contains the highest proportion of Leu residues, at 12%, followed by 6.7% Glu and 6.7% Pro. The total positive and negative charges are 63 and 60, respectively. In addition, the total average hydrophilicity is −0.228. The CYP2E1 protein has no obvious hydrophobic region, and the number of hydrophilic amino acid residues is more than that of hydrophobic residues; therefore, it can be inferred that the protein is hydrophilic, and the results of a protein hydropathy analysis by ProScale also indicate that CYP2E1 is a hydrophilic protein (Appendix A). To further analyze the secondary structure and hydropathy characteristics of the CYP2E1 protein, the sequence was uploaded to the NCBI website for online prediction. The secondary structure of the CYP2E1 protein consists of α-helices, random coils, extended chains, and β-turns (Appendix A). Among them, α-helices and random coils are the main components of the secondary structure of the CYP2E1 protein, accounting for 46.45% and 36.31%, respectively, while extended chains and β-turns account for 12.17% and 5.07%. Then, the tertiary structure of the CYP2E1 protein was predicted using Swiss-Model online server (Appendix A).

### 2.3. Prediction of Transmembrane Regions, Signal Peptides, Subcellular Localization, and Phosphorylation Sites of CYP2E1 Protein

Using DeepTMHMM online server for the prediction of transmembrane regions of the *L. yarkandensis* CYP2E1 protein, the results showed that the *L. yarkandensis* CYP2E1 protein has one transmembrane helix (TM helix) from amino acid position 6 to 20. The N-terminal (1–5) is located outside the membrane (Outside), the transmembrane helix (6–20) is in the membrane (Membrane), and the C-terminal (21–493) is inside the membrane (Inside), and the presence of a signal peptide may be in a hydrophobic region at the N-terminal, which helps guide it to the endoplasmic reticulum for proper folding and localization (Appendix A). Using the online tool DeepLoc—2.1 for subcellular localization prediction, the results showed that the protein is most likely to be located in the endoplasmic reticulum membrane with a probability of 89.35% (Appendix A). By predicting its phosphorylation sites with NetPhos3.1, the results showed that the *L. yarkandensis* CYP2E1 protein has a total of 40 phosphorylation sites, including 18 serine (S) sites, 15 threonine (T) sites, and seven tyrosine (Y) sites (Appendix A).

### 2.4. Heterologous Expression of CYP2E1 Gene in Yeast

This study involved the construction of the pYES2-*CYP2E1* yeast expression vector to analyze the growth resistance of the INVSc1 strain. Double digestion with EcoRI and XboI confirmed the successful construction of the pYES2-*CYP2E1* recombinant plasmid. Western blot analysis further validated the expression of the construct in INVSc1 yeast cells (Figure 3B).

### 2.5. Heterologous Expression of CYP2E1 Gene Enhances Yeast Tolerance to Drought and Its Effects on Antioxidant Enzyme Activity and Oxidative Stress Markers

After treating yeast cells with mannitol to simulate drought conditions, we observed a general decrease in cell survival rates (Figure 4A). Specifically, in the control group without mannitol treatment, a few colonies formed even at a dilution of 10⁻^4^. As the mannitol concentration increased to 0.75 M, the yeast strain INVSc1-pYES2, which did not express the *CYP2E1* gene, failed to form any observable colonies at the same dilution, while the strain expressing *CYP2E1* (INVSc1-pYES2*-CYP2E1*) formed a few colonies. When the concentration of mannitol was further increased to 1 M, INVSc1-pYES2 could only form two single colonies at a dilution of 10⁻^3^ and failed to form any colonies at a dilution of 10⁻^4^. In contrast, INVSc1-pYES2-*CYP2E1* was able to form seven single colonies even at a dilution of 10⁻^3^ and still managed to form one single colony at a dilution of 10⁻^4^. These results indicate that the expression of *CYP2E1* significantly enhanced the yeast’s tolerance to drought conditions.

To further explore the physiological effects of mannitol treatment on yeast cells, we subsequently measured the activities of superoxide dismutase (SOD) (Figure 4B) and catalase (CAT) (Figure 4C), as well as the levels of reactive oxygen species (ROS) (Figure 4D), protein oxidation (Pro) (Figure 4E), and malondialdehyde (MDA) (Figure 4F). In this study, compared to the control group, the transgenic yeast expressing the *CYP2E1* gene demonstrated significant physiological differences when subjected to mannitol treatment; the activities of both SOD and CAT were enhanced, suggesting that CYP2E1 may strengthen the yeast’s antioxidant defense mechanisms. Meanwhile, the transgenic yeast exhibited lower MDA levels, which may reflect better protection of the cell membrane and reduced lipid peroxidation damage. Although the ROS levels were higher in the transgenic yeast, this could be attributed to the increased metabolic activity and superoxide anion production associated with CYP2E1. Furthermore, the higher Pro levels in the transgenic yeast may not only indicate enhanced protein stability but also involve other aspects of osmotic regulation and intracellular homeostasis. In terms of common trends, both groups of yeast showed an initial increase followed by a decrease in SOD and CAT activities, a continuous rise in ROS levels, and a sharp increase in MDA levels at 0.75 M mannitol concentration, revealing fluctuations in the antioxidant defense system and challenges to cell membrane stability under high osmotic stress. These findings indicate that at the 0.75 M mannitol concentration, the yeast’s antioxidant system and cell membrane stability face severe trials, and the expression of the *CYP2E1* gene may enhance the yeast’s tolerance to these environmental stresses, thereby improving its survival capabilities.

### 2.6. Heterologous Expression of CYP2E1 Gene Enhances Yeast Tolerance to Saline Conditions and Its Effects on Antioxidant Enzyme Activity and Oxidative Stress Markers

After treating yeast cells with NaCl to simulate saline conditions, this study found that the survival rate of yeast cells generally decreased (Figure 5A). Specifically, in the control group without added NaCl, a few colonies still formed even at a dilution of 10^−4^. However, when the NaCl concentration increased to 1 M, no colony growth was observed in either the transgenic group or the empty vector group at the same dilution. At a dilution of 10^−3^, 13 single colonies were observed in the transgenic group, while nine were observed in the empty vector group. Under 3 M NaCl treatment, only one single colony was observed in the empty vector group at a dilution of 10^−3^, while the transgenic group observed seven. These results indicate that the heterologous expression of the *CYP2E1* gene significantly enhanced the yeast cells’ tolerance to saline environments.

To further explore the impact of NaCl treatment on yeast cell physiological indicators, this study additionally measured the activities of SOD (Figure 5B), CAT (Figure 5C), and the levels of ROS (Figure 5D), Pro (Figure 5E), and MDA (Figure 5F) in Saccharomyces cerevisiae. The transgenic yeast under salt stress showed higher SOD and CAT activities, which may suggest that CYP2E1 plays a role in enhancing the yeast’s defense against oxidative stress. Moreover, the transgenic yeast under salt stress also exhibited higher Pro and ROS levels, as well as lower MDA levels. In terms of common trends, SOD activity initially increased and then gradually decreased, while CAT activity generally showed a downward trend. At 0–1 M NaCl concentrations, ROS and MDA levels slowly and continuously increased, indicating a continuous accumulation of oxidative stress within the cells. When the NaCl concentration exceeded 1 M, ROS and MDA levels rose exponentially, indicating severe oxidative stress damage to the cells under high osmotic pressure stress. The gradual decrease in Pro may reflect an increase in protein damage and degradation within the cells. These results indicate that under 1 M NaCl treatment, the antioxidant defense system of yeast cells experienced significant fluctuations. It is evident that under extreme salt stress, the antioxidant system and membrane stability of yeast cells face severe challenges, and the expression of the *CYP2E1* gene may enhance the yeast cells’ tolerance to these environmental stresses to some extent.

### 2.7. Heterologous Expression of CYP2E1 Gene Enhances Yeast Tolerance to High Temperatures

After treatment at temperatures ranging from 30 °C to 55 °C, the survival rate of yeast cells generally decreased (Figure 6A). Specifically, under conditions at 30 °C, a large number of colonies were observed, even at a dilution of 10^−5^. However, when the temperature was raised to 45 °C, the number of colonies significantly decreased for both yeast strains at the same dilution; at a dilution of 10^−4^, no colonies were observed for the empty vector group, while the transgenic group still managed to form a single colony. Under treatment at 55 °C, no colonies were found for the empty vector group, while the transgenic strain was able to form ten single colonies even without dilution. These results indicate that the heterologous expression of the *CYP2E1* gene significantly enhanced the yeast cells’ tolerance to high-temperature environments.

To further explore the impact of different temperature treatments on yeast cell physiological indicators, researchers measured the activities of superoxide dismutase (SOD, Figure 6B), catalase (CAT, Figure 6C), and the levels of reactive oxygen species (ROS, Figure 6D), protein oxidation (Pro, Figure 6E), and malondialdehyde (MDA, Figure 6F) in Saccharomyces cerevisiae. Similarly to the results under drought and salt stress conditions, the transgenic yeast under high-temperature stress also showed higher SOD and CAT activities, which may suggest that CYP2E1 plays a significant role in enhancing the yeast’s defense against oxidative stress. Additionally, the transgenic yeast under high-temperature stress also exhibited higher ROS levels and lower MDA levels, consistent with observations under drought conditions. Interestingly, the Pro levels in the transgenic yeast group were initially higher and then became similar to those in the empty vector group, reflecting the changing demands for proline under high-temperature stress. Concurrently, the rapid death rate in the empty vector group led to a swift increase in free proline, which showed a trend of surpassing that of the transgenic group.

In terms of common trends, under treatment at 45 °C, the antioxidant defense systems of both yeast strains showed significant fluctuations. SOD activity initially increased and then gradually decreased, which may be due to the cells enhancing the production of antioxidant enzymes to cope with increased ROS, but as time progressed and cell damage intensified, SOD activity gradually decreased. Meanwhile, CAT activity generally showed a downward trend. With treatments from 30 °C to 45 °C, ROS levels slowly and continuously increased, indicating a continuous accumulation of oxidative stress within the cells. When the temperature reached between 45 °C and 55 °C, ROS levels surged dramatically, indicating severe oxidative stress damage to the cells under high-temperature stress. MDA levels slowly and continuously increased under treatments from 30 °C to 40 °C, further indicating the continuous accumulation of oxidative stress within the cells. When the temperature reached between 40 °C and 55 °C, MDA levels surged dramatically, showing severe oxidative stress damage to the cells under high-temperature stress. Pro activity generally increased, especially a noticeable rise at 50–55 °C, indicating that the cells initially produced a large amount of proline to cope with the stress, but as the temperature became too high, it may lead to cell death, and the combined proline was released due to the high-temperature effect, leading to a continuous increase in Pro levels in the detection results. These results indicate that around 45 °C, the antioxidant defense system of yeast cells experienced significant fluctuations. When facing high-temperature stress, the antioxidant system and membrane stability of yeast cells face severe challenges, and the expression of the *CYP2E1* gene may enhance the yeast cells’ tolerance to these environmental stresses to some extent.

### 2.8. Impact of L. yarkandensis CYP2E1 Gene on Stress Resistance of Saccharomyces cerevisiae INVSc1

This study successfully cloned the *CYP2E1* gene from the kidneys of *L. yarkandensis* and transformed it into Saccharomyces cerevisiae INVSc1. Under stress conditions such as drought, salinity, and high temperatures, the yeast strain with the *CYP2E1* gene exhibited a higher survival rate compared to the control without the *CYP2E1* gene. Additionally, the expression of this gene enhanced the activity of SOD and CAT in yeast cells, suggesting that CYP2E1 may bolster yeast stress resistance by increasing its antioxidant capacity. Although *CYP2E1* expression led to increased ROS levels, this was accompanied by elevated Pro and MDA levels, potentially reflecting an adaptive mechanism in yeast cells to enhance overall survival under stress by modulating redox balance. These findings indicate that *CYP2E1* expression might improve yeast survival under stress conditions by regulating the redox equilibrium within yeast cells (Figure 7).

## 3. Discussion

This study successfully obtained the complete CDS region nucleotide sequence of the *CYP2E1* gene from the kidney of *L. yarkandensis* through gene cloning techniques, which is 1482 base pairs long and encodes 493 amino acids. Although the amino acid sequence identity between the *L. yarkandensis* CYP2E1 and that of *O. cuniculus* reaches 99%, a critical alignment analysis revealed differences at 39 nucleotide positions, with 13 of these differences leading to amino acid changes. Specifically, for instance, the L/F substitution at position 19, although classified as a conservative replacement, enhances the hydrophobicity of the protein. In contrast, the S/F substitution at position 128 is a non-conservative replacement, which alters the amino acid from hydrogen bond-forming to non-hydrogen bond-forming, thereby affecting the protein’s hydrogen bond network. Furthermore, the C/S substitution at position 268, also a non-conservative replacement, disrupts an existing disulfide bond. Similarly, the S/P substitution at position 462, which involves a transition from a hydrogen bond-forming amino acid to a cyclic amino acid that cannot form hydrogen bonds, may have profound effects on the bending of the protein chain and its secondary structure. These specific sequence variations may be crucial for the adaptation of *L. yarkandensis* to extremely arid environments. Phylogenetic tree analysis further confirmed the high homology between *L. yarkandensis* and *O. cuniculus* while also highlighting the conservation of the *CYP2E1* gene during the evolutionary process, which may be related to its key role in metabolism and detoxification processes. These findings emphasize that even within highly conserved genes, minor sequence changes can have a significant impact on the adaptability of a species.

Bioinformatics tools were used to predict the physicochemical properties and structure of the *L. yarkandensis* CYP2E1 protein. The results of this study show that the molecular weight of the CYP2E1 protein is about 57 KDa, and the theoretical isoelectric point is 8.37, which is similar to the physicochemical properties of known CYP2E1 proteins. In addition, the secondary structure of the CYP2E1 protein is mainly composed of α-helices and random coils, which is consistent with its function as a metabolic enzyme. Online tools were used to predict the transmembrane regions and subcellular localization of the *L. yarkandensis* CYP2E1 protein. This predicted result is consistent with its characteristic as a membrane-bound enzyme. Although CYP2E1 itself does not contain a typical transmembrane domain, it can bind and anchor to the endoplasmic reticulum membrane through its N-terminal signal peptide and C-terminal anchoring sequence, thus explaining the phenomenon of the predicted transmembrane helix. In addition, the CYP2E1 protein is rich in phosphorylation sites, suggesting that the protein may play a central role in various biological functions within the cell. These functions include but are not limited to the metabolism of drugs and toxins, the regulation of cellular stress responses, and processes related to the occurrence and development of diseases. Therefore, the phosphorylation status of CYP2E1 may be crucial for maintaining cellular homeostasis and adapting to environmental changes [26].

Additionally, CYP2E1, as an important metabolic enzyme, plays a significant role in metabolism within animals. For instance, CYP2E1 plays a central role in the metabolism of 1,4-dioxane in animal livers and is one of the key enzymes involved in the processing of such water pollutants [27]. Triphenyl phosphate (TPhP) and its mono-hydroxylated metabolites are more readily metabolized by the CYP2E1 enzyme in mammals [28]. Due to the increasing industrialization in camel-rearing regions, driven by modernization and the expanding industrial revolution, camels’ CYP2E1 enzyme has evolved to effectively metabolize small toxins such as aniline, benzene, catechol, amides, butadiene, toluene, and acrylamide [29]. APAP undergoes hydroxylation via CYP2E1 to form NAPQI, which then, under the action of glutathione, is converted into forms that are more easily excreted from the body, thereby reducing the accumulation and potential toxicity of APAP within the body [30]. Studies have found that inhalational anesthetics can induce the expression of CYP2E1, thereby accelerating drug metabolism [31].

CYP2E1, as an important metabolic enzyme, has been studied for its ability to enhance the metabolic capacity of plants to organic pollutants when heterologously expressed in plants. For example, *CYP2E1* transgenic poplar initiated the TCE metabolic pathway, and the accumulation of TCE metabolites triggered the high expression of glutathione S-transferases, glycosyltransferases, and ABC transporters [32]. Compared with wild-type Ardisia pusilla, *CYP2E1* transgenic *Ardisia pusilla* showed an increase in the removal capacity of external toluene at different time points by 5.1 to 6.1 times [33]. *Petunia hybrida* containing the *CYP2E1* gene can effectively remove benzene and toluene pollutants and improve resistance to formaldehyde [34,35]. The expression of human cytochrome P4502E1 in *Nicotiana tabacum* enhanced the tolerance and remediation of γ-hexachlorocyclohexane [36]. Although the application of the CYP2E1 enzyme in plants is relatively widespread, its application in microorganisms, which are major handlers of environmental pollutants, is relatively less.

In the yeast expression system, the heterologous expression of the *CYP2E1* gene significantly improved the tolerance of yeast cells to drought, salinity, and high-temperature environments (Figure 7). In this experiment, compared to the yeast transformed with the empty vector, the genetically modified yeast exhibited different physiological responses when facing adverse environmental conditions. Specifically, genetically modified yeast showed higher levels of reactive oxygen species (ROS) but lower levels of MDA, as well as higher levels of SOD, CAT, and Pro. These results suggest that the expression of the *CYP2E1* gene may enhance the tolerance of yeast to oxidative stress and osmotic pressure changes. The higher levels of ROS may originate from the enhanced metabolic activity of CYP2E1, leading to an increase in the production of superoxide anions. However, the lower levels of MDA indicate that genetically modified yeast more effectively prevented lipid peroxidation and protected the cell membrane from oxidative damage, which may be due to the enhancement of its antioxidant defense system. The increase in SOD and CAT activity suggests that the expression of CYP2E1 may enhance the resistance of yeast to oxidative stress by accelerating the removal of ROS to protect cells. The increase in proline content may help maintain the stability of the intracellular environment and protect cells from damage caused by various environmental stresses. In mammals, the expression of *CYP2E1* can activate the Nrf2 signaling pathway. Nrf2 is a transcription factor that can regulate the expression of important antioxidant and Phase II detoxification genes [7]. In yeast expressing *CYP2E1*, the expression of antioxidant enzymes and protective molecules such as proline may be upregulated to cope with increased oxidative stress. Therefore, the expression of the *CYP2E1* gene may improve the tolerance of yeast to drought and salt stress by enhancing the antioxidant defense system and osmotic pressure regulation mechanism of yeast cells.

It is noteworthy that proline is an important non-protein amino acid that acts as an osmotic regulator in plants. It can accumulate under adverse conditions to maintain the osmotic balance between the inside and outside of cells, thereby enhancing the resistance of plants. In addition to participating in osmoregulation, the accumulation of proline in plants is also closely related to its antioxidant capacity and protein structural stability. However, research on the mechanism of proline action and its accumulation patterns in yeast is still in its infancy. Therefore, monitoring the content of proline in the yeast expression system not only provides a new perspective for evaluating yeast resistance but also opens up a new direction for in-depth exploration in this field.

In this study, we observed a very interesting phenomenon, that is, under specific stress conditions, yeast cells (INVSc1-pYES2-*CYP2E1* and INVSc1-pYES2) showed a significant physiological fluctuation point, which we defined as the “Yeast Oxidative Stress Lethal Threshold (Yeast OSLT)”. This concept refers to the critical point under certain environmental stresses where the yeast cell’s antioxidant defense system cannot maintain cellular homeostasis, leading to massive cell death. The mechanism of this threshold includes excessive oxidative stress causing DNA damage, protein oxidation, and lipid peroxidation, ultimately leading to cell death [37]. Yeast cells have a complex antioxidant defense system, including superoxide dismutase (SOD), catalase (CAT), etc., which work together to eliminate ROS and protect cells from oxidative damage [38]. *CYP2E1*, expressed in the kidney, may be involved in yeast metabolism and detoxification processes, thereby protecting yeast cells to some extent.

This study offers molecular insights into the survival strategies of *L. yarkandensis* in extreme environments, establishing a preliminary foundation for further exploration of the *CYP2E1* gene’s role in stress adaptation. The *CYP2E1* gene is likely to play a significant role in *L. yarkandensis*’s ability to adapt to extreme conditions, and it may also hold potential applications in the fields of biodiversity conservation and biotechnology. While it is hypothesized that the *CYP2E1* gene might play a similar role in other species adapted to extreme environments, this hypothesis requires further research for confirmation. By investigating the sequence variations, spatial structures, expression patterns, and biological functions of the *CYP2E1* gene across different species, we may be able to develop novel conservation strategies to assist these species in adapting more effectively to environmental changes.

Furthermore, the potential application of the *CYP2E1* gene in plant stress resistance breeding and environmental remediation warrants additional exploration. Introducing the *CYP2E1* gene from *L. yarkandensis* into plants could potentially enhance their resistance to adversities such as drought, salinity, and high temperatures, which may consequently lead to increased crop yields and improved quality. Moreover, harnessing the metabolic capabilities of the *L. yarkandensis CYP2E1* gene could significantly boost the ability of microorganisms to degrade environmental pollutants, including trichloroethylene, chloroform, and benzene, as well as their survival under adverse conditions, offering innovative solutions for environmental remediation. These findings not only deepen our understanding of biological adaptability but also pave the way for new avenues in biotechnological applications, with a particular emphasis on the *CYP2E1* gene from *L. yarkandensis*.

## 4. Materials and Methods

### 4.1. Animals and Tissues

The experimental animals were adult male *O. cuniculus*, approximately 8 months old. All experimental animals used in this study were raised and utilized in accordance with the ARRIVE guidelines. The use of experimental animals and procedures were approved by the Scientific Research Office of Tarim University in Xinjiang Uygur Autonomous Region, with the approval number 2022015. The care and use of experimental animals strictly adhered to local animal welfare laws, guidelines, and policies. Six *L. yarkandensis* were collected from Shawan County in the Aksu region, northwest of *L. yarkandensis*. Six adult *O. cuniculus* were obtained from the Animal Experimental Station of Tarim University. Animals were determined to be adult based on a skull length greater than 75.50 mm. All selected animals were in good health.

Husbandry and diet: Prior to the study, *L. yarkandensis* were acclimated for one week with a diet consisting of local grasses and freshwater from the same region to simulate their natural habitat. *O. cuniculus* were kept under standard husbandry conditions. Both types of rabbits were housed individually in spacious, well-ventilated cages with constant temperature and humidity to ensure their comfort. They had free access to food and water, and their intake was monitored daily to assess their health status and dietary adaptation. After the acclimation period, rabbits were anesthetized with 3% pentobarbital sodium (5 mg/kg) for kidney dissection. The left kidney tissue was collected, cut into small pieces, and then placed into cryovials containing RNA preservation solution for subsequent RNA analysis.

### 4.2. Cloning of L. yarkandensis CYP2E1

#### 4.2.1. RNA Extraction and cDNA Synthesis

Total RNA was extracted using the TransZol Up Plus RNA ki (Transgen, Beijing, China) and stored at −80 °C. The integrity and concentration of the total RNA were determined using 0.8% agarose gel electrophoresis to ensure they met the trial standards. EasyScript^®^ One-sep gDNA Removal and cDNA Synthesis SuperMix (Transgen, Beijing, China) was used for cDNA synthesis, incubated at 42 °C for 15 min, inactivated at 85 °C for 5 s, and stored at −20 °C.

#### 4.2.2. Primer Design and Synthesis

Based on the *CYP2E1* gene sequence of *O. cuniculus* in the NCBI (GenBank No. XM_002718772.4), primers for *CYP2E1* gene cloning with enzyme cutting sites were designed using Premier5.0 software. Primer sequences are shown in Table 1. Primers were synthesized by Shanghai Shenggong Bioengineering (Shanghai, China) Co., Ltd.

#### 4.2.3. Cl4.2.3 Cloning and Sequencing of the CYP2E1 Gene CDS Region

PCR amplification was performed using the obtained cDNA as a template, with the expected product size of 1541 base pairs. A 20-microliter PCR reaction system was used, with 1 microliter of cDNA template, 0.4 microliters each of upstream and downstream primers (10 micromolar/L), 10 microliters of Primer Star Max, and 8.2 microliters of ddH2O. The PCR amplification program was as follows: initial denaturation at 95 °C for 3 min; denaturation at 94 °C for 10 s; annealing at 55.6 °C for 30 s; extension at 72 °C for 1 min and 30 s; and a final hold at 4 °C for 5 min. PCR products were separated by 0.8% agarose gel electrophoresis, and the target bands were recovered. The target fragments were ligated with the pMD-19T vector overnight at 16 °C. Ligated products were transformed into ampicillin-resistant DH5α competent cells, and the plates were incubated overnight in a 37 °C incubator. Positive clones were selected based on blue and white spots, picked in ampicillin-resistant LB liquid medium, and bacterial solutions were identified by PCR, followed by sequencing at Shanghai Shenggong Bioengineering Co., Ltd.

### 4.3. Structural and Functional Analysis of CYP2E1 Sequence

DNA and protein sequences were analyzed using online BLAST software (https://blast.ncbi.nlm.nih.gov/Blast.cgi, accessed on 20 August 2022). The ORFfinder program (https://www.ncbi.nlm.nih.gov/orffinder, accessed on 19 August 2022) was used to predict open reading frames. The ExPASy ProtParam tool (http://web.expasy.org/protparam, accessed on 24 August 2022) was utilized to predict and calculate the theoretical isoelectric point (pI), molecular weight (MW), and amino acid composition of the proteins. ClustalW1.83 (https://www.genome.jp/tools-bin/clustalw, accessed on 30 August 2022) was utilized to perform alignments of both nucleotide sequences and protein sequences of the *CYP2E1* gene under default settings. The alignment output was used for analysis with DNAMAN8.0 software (Lynnon BioSoft, Quebec, QC, Canada) and also for generating a phylogenetic tree based on the neighbor-joining (NJ) method with 1000 bootstrap replicates. The GOR IV secondary structure prediction method (https://npsa-prabi.ibcp.fr/cgi-bin/npsa_automat.pl?page=npsa_gor4.html, accessed on 18 December 2022) was used to predict the secondary structure of the deduced amino acid sequences. The tertiary structure of the CYP2E1 protein was predicted using the Swiss-Model online server that is accessible through the link provided (https://swissmodel.expasy.org, accessed on 18 December 2022). Based on the complete protein sequences, the SMART database (http://smart.emblheidelberg.de, accessed on 18 March 2023) and the transmembrane prediction server TMHMM 2.0 (http://www.cbs.dtu.dk/services, accessed on 18 March 2023) were used to predict transmembrane domains (TMDs), and the NetPhos 3.1 server (https://services.healthtech.dtu.dk/services/NetPhos-3.1/) was used to predict phosphorylation sites in eukaryotic proteins.

### 4.4. Yeast Transformation

The *CYP2E1* gene cDNA sequence was used as a template for PCR amplification with primers pYES2-*CYP2E1*-F and pYES2-*CYP2E1*-R (Table 1). The target fragment and pYES2 empty vector were double digested with Pspx I and Not I (Takara, Beijing, China) at 37 °C for 30 min; T4 DNA ligase (Takara, Beijing, China) was used for overnight ligation of the target and vector fragments. The recombinant vector was transformed into E.coli DH5α competent cells using the heat shock method, and after plating, incubated at 37 °C for 16 h. Single colonies were picked for colony PCR validation, and positive clones were extracted for plasmid double-digestion testing to obtain the positive clone plasmid pYES2-*CYP2E1*. Carrier DNA was inserted into a 95 °C metal bath for 5 min, then quickly placed on ice. A total of 100 μL of ice-melted INVSc1 competent cells (WeiDi, Shanghai, China) were taken, and 2–5 pg of pre-cooled target plasmid pYES2-*CYP2E1* and 10 μL of pre-treated carrier DNA were added, followed by 500 μL of PEG/LiAc and mixing by flicking. The mixture was incubated at 30 °C for 30 min (flicking 8 times at 15 min). The tube was placed in a 42 °C water bath for 15 min (flicking 6–8 times at 7.5 min). After centrifugation at 5000 rpm for 40 s, the supernatant was taken, resuspended with 400 μL of ddH2O, centrifuged for 30 s, and the supernatant was discarded. The pellet was resuspended with 50 μL of ddH2O, spread on SC-u/2% (*w*/*v*) glucose plates, and incubated at 29 °C for 48–96 h. Single clones were picked for PCR detection. The empty vector pYES2 was introduced into INVSc1 competent cells using the same method.

### 4.5. Validation of Heterologous Expression of CYP2E1

Monoclonal yeast cells INVSc1-pYES2-*CYP2E1* and monoclonal yeast cells INVSc1-pYES2 were inoculated into 10 mL of SC-U/2% (*w*/*v*) glucose liquid medium and incubated at 30 °C with shaking for 20 h. The OD600 of the yeast culture was measured, and the culture was then inoculated into 10 mL of induction expression medium (SC-U/2% (*w*/*v*) galactose liquid medium), adjusting the OD600 to 0.4 and incubating at 30 °C with shaking for 36 h to promote the expression of the foreign gene. Recombinant proteins were extracted using a yeast protein extraction kit (Solarbio, Beijing, China) for later use. The concentration of the extracted proteins was determined using a BCA assay kit, and the proteins were adjusted to a uniform concentration, mixed with loading buffer and boiled in water at over 98 °C for 5 min. SDS/PAGE electrophoresis technology was employed, followed by transferring the proteins onto a membrane at a constant current of 200 mA for 90 min. The membrane was blocked with 5% skim milk at room temperature for 2 h. The primary antibodies (CYP2E1 at 1:3000 and GAPDH at 1:10,000 dilutions) and secondary antibody (horseradish peroxidase (HRP)-conjugated IgG antibody at 1:5000 dilution) were incubated successively.

### 4.6. Stress Resistance Testing of Transgenic Yeast Cells

In the experiment, a uniform method was employed to treat yeast cells to study their adaptability to various environmental stresses. Initially, 0.25, 0.5, 0.75, and 1 mol/L mannitol solutions were prepared to simulate drought conditions; 0.5, 1, 2, and 3 mol/L NaCl solutions were prepared with 0.01 M PBS solution to simulate saline conditions, with 0.01 M PBS solution serving as a control, and each experimental group was replicated three times. Monoclonal yeast cells after protein induction were centrifuged to discard the supernatant after adjusting the OD600 to 1, then resuspended with equal volumes of different concentrations of NaCl or mannitol solutions and 0.01 M PBS, divided into groups with recombinant plasmids and empty plasmids. For the saline and drought groups, the cells were cultured at 30 °C for 16 h, whereas the heat treatment group was exposed to temperatures of 30 °C, 35 °C, 40 °C, 45 °C, 50 °C, and 55 °C for 1 h. The treated cell suspensions were serially diluted to 10^0^, 10⁻^1^, 10⁻^2^, 10⁻^3^, 10⁻^4^, and 10⁻^5^, and then 1 μL of each dilution was spotted onto SD-Ura agar medium. After diluting the cell suspensions and spotting them onto SD-Ura agar medium, they were incubated at 30 °C for 72 h to observe colony growth, thereby assessing the impact of different concentrations of solutions and temperatures on the growth of yeast cells.

### 4.7. Detection of Intracellular SOD, PRO, CAT, MDA, and ROS Levels

After stress treatment in “4.6”, 1 mL of cell suspension was taken from each group, with three replicates per group, and the activities of antioxidant enzymes SOD and CAT, as well as the levels of ROS, Pro, and MDA, were detected according to the instructions provided with the assay kits (Solarbio, Beijing, China).

### 4.8. Statistical Analysis

All data in this experiment were expressed as mean ± standard deviation (X ± S) and GraphPad software (version 10.1.2, San Diego, CA, USA) was used for significance analysis. A *t*-test was used to compare the two independent samples and *p* ≤ 0.05 was considered statistically significant. In all graphs, *p* ≤ 0.05 was marked as *, *p* ≤ 0.01 was marked as **, *p* ≤ 0.001 was marked as ***, and n ≥ 3.

## 5. Conclusions

In this study, the *CYP2E1* gene from the kidneys of the *L. yarkandensis* hare was identified. The study indicates that the protein encoded by this gene is highly homologous to that of other species, but even the *O. cuniculus*, which has the highest sequence similarity, still exhibits 13 amino acid differences. Heterologous expression experiments conducted in a yeast expression system have shown that the *CYP2E1* gene can significantly enhance the tolerance of yeast cells to drought, saline, and high-temperature environments, which may be related to the increased activity of antioxidant enzymes and the reduced levels of oxidative stress markers. Additionally, the study observed a physiological threshold in yeast cells under specific stress conditions, known as the “Yeast Oxidative Stress Lethal Threshold” (Yeast OSLT). The findings provide molecular insights into how the *L. yarkandensis* hare survives in extreme environments and offer significant scientific value for the study of biodiversity conservation and ecosystem stability. Future research may further explore the expression patterns of CYP2E1 in other tissues of the *L. yarkandensis* hare and its role in environmental adaptation, as well as the functional study of CYP2E1 in other species adapted to extreme conditions. This could reveal the universal role of CYP2E1 in biological adaptability and provide additional scientific evidence for the conservation of biodiversity and the maintenance of ecosystem stability.

## Figures and Tables

**Figure 1 ijms-26-00453-f001:**
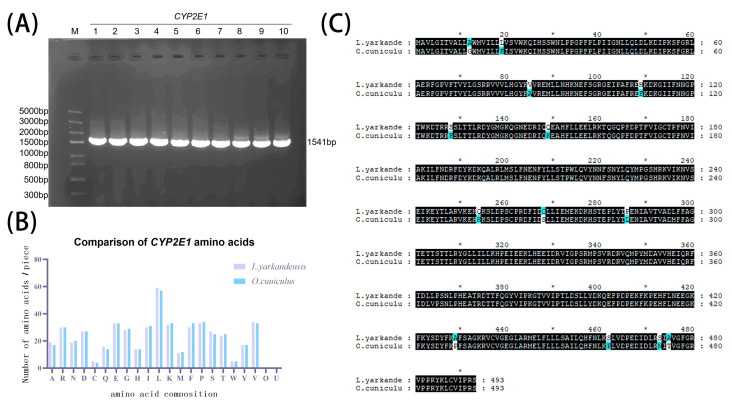
Cloning of the *CYP2E1* gene from *L. yarkandensis* and a sequence comparison with the amino acids of *O. cuniculus*. (**A**) Agarose gel electrophoresis detection results of *L. yarkandensis CYP2E1* gene PCR products. (**B**) Comparison of individual amino acid numbers between *L. yarkandensis* and *O. cuniculus* CYP2E1 proteins. (**C**) Amino acid sequence alignment of *L. yarkandensis* and *O. cuniculus* CYP2E1 proteins.

**Figure 2 ijms-26-00453-f002:**
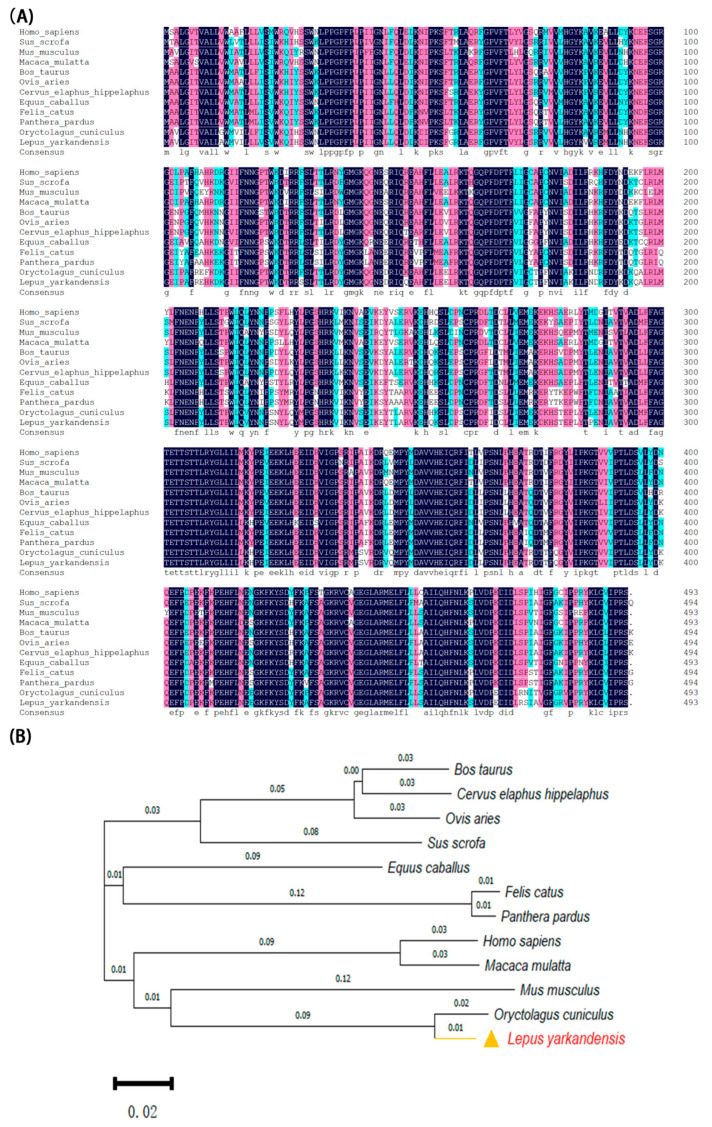
Alignment and phylogenetic analysis of CYP2E1. (**A**) Multiple sequence alignment analysis of CYP2E1. (**B**) Phylogenetic tree of CYP2E1 protein and homologous sequences (neighbor joining, NJ). The values on the branches of the phylogenetic tree are bootstrap values. Abbreviated names and GenBank accession numbers are as follows: *Bos taurus* (AAY83882.1), *Cervus elaphus hippelaphus* (OWK08011.1), *Ovis aries* (ADZ11094.1), *Sus scrofa* (NP_999586.1), *Felis catus* (NP_001041475.1), *Panthera pardus* (XP_019287846.2), *Homo sapiens* (NP_000764.1), *Macaca mulatta* (AAT49269.1), *Mus musculus* (NP_067257.1), and *O. cuniculus* (XP_002718818.1).

**Figure 3 ijms-26-00453-f003:**
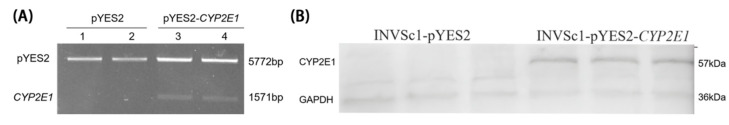
Heterologous expression of *CYP2E1* gene in yeast. (**A**) Double digestion identification of pYES2 and pYES2-*CYP2E1*. (**B**) Western blotting detection of CYP2E1 protein expression levels in INVSc1-pYES2 and INVSc1-pYES2-*CYP2E1*.

**Figure 4 ijms-26-00453-f004:**
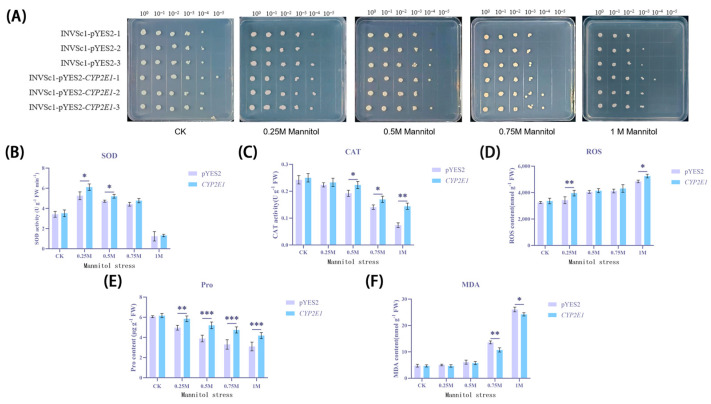
Physiological effects of heterologous expression of *CYP2E1* gene on yeast cells under mannitol stress. (**A**) Colony formation of yeast cells (INVSc1-pYES2 and INVSc1-pYES2-*CYP2E1*) under different concentrations of mannitol treatment. (**B**) Changes in SOD activity. (**C**) Changes in CAT activity. (**D**) Changes in ROS levels. (**E**) Changes in Pro levels. (**F**) Changes in MDA levels. * *p* < 0.05; ** *p* < 0.01; *** *p* < 0.001.

**Figure 5 ijms-26-00453-f005:**
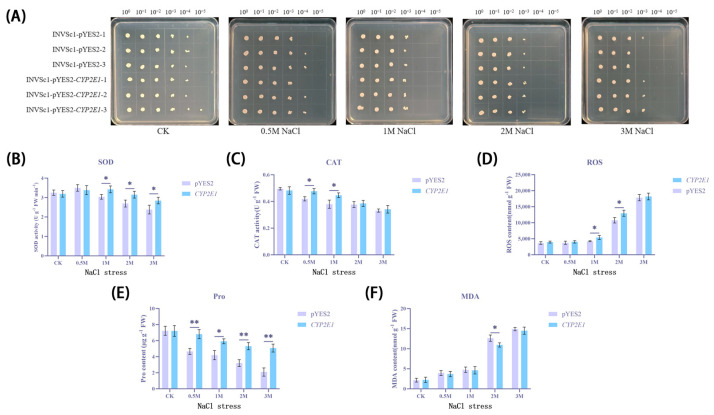
Physiological effects of heterologous expression of *CYP2E1* gene on yeast cells under NaCl stress. (**A**) Colony formation of INVSc1-pYES2 and INVSc1-pYES2-*CYP2E1* under different concentrations of NaCl treatment. (**B**) Changes in SOD activity. (**C**) Changes in CAT activity. (**D**) Changes in ROS levels. (**E**) Changes in Pro levels. (**F**) Changes in MDA levels. * *p* < 0.05; ** *p* < 0.01.

**Figure 6 ijms-26-00453-f006:**
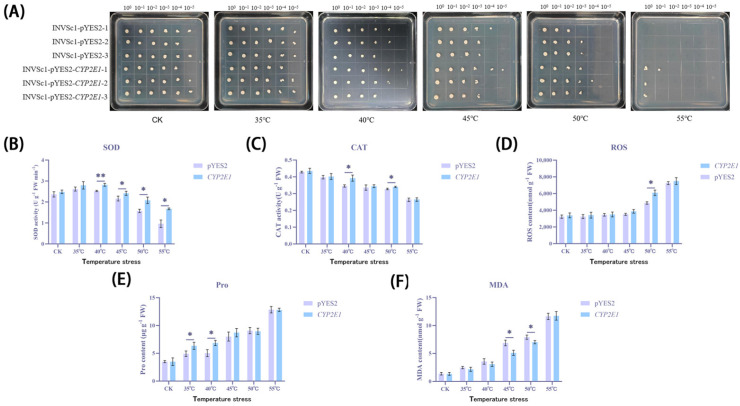
Physiological effects of heterologous expression of *CYP2E1* gene on yeast cells under high-temperature stress. (**A**) Colony formation of INVSc1-pYES2 and INVSc1-pYES2-*CYP2E1* under different temperature treatments. (**B**) Changes in SOD activity. (**C**) Changes in CAT activity. (**D**) Changes in ROS levels. (**E**) Changes in Pro levels. (**F**) Changes in MDA levels. * *p* < 0.05; ** *p* < 0.01.

**Figure 7 ijms-26-00453-f007:**
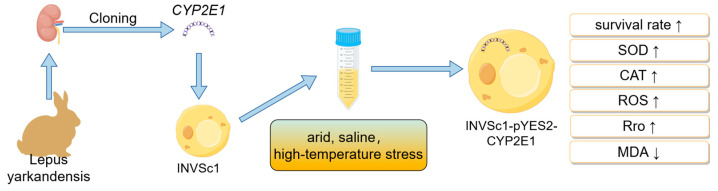
Visualization of stress response steps in *CYP2E1*-transformed yeast.

**Table 1 ijms-26-00453-t001:** Sequences of target genes and internal reference primers.

Gene	Direction	Primer Sequence (5′-3′)	Note
*CYP2E1*	F	ATGGCTGTTCTGGGCATCACCG	Coding sequence amplifcation
	R	TTACGAGCGGGGAATGACACAGAGT	
pYES2*-CYP2E1*	F	GCCTCGAGCTGCCACCATGGCTGTTCTGGGCATCAC	Restriction enzyme site primer
	R	TGCTCTAGAGCATTACGAGCGGGGAATGACACAGA	

## Data Availability

The data presented in this study are available upon request from the corresponding author.

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
