# Peer review of "The Functional Identification of the CYP2E1 Gene in the Kidney of Lepus yarkandensis"

_ijms, 2025, doi:10.3390/ijms26020453_

Round 1
Reviewer 1 Report
Comments and Suggestions for Authors
This piece of work contains some data attempting to identify the function of the renal cytochrome P450 2E1 (CYP2E1) gene in L. yarkandensis. This animal is a Chinese hare with ability to survive in extremely arid conditions. CYP2E1 is a member of the cytochrome P450 family with important roles in the metabolism of various endogenous and exogenous compounds. The manuscript contains an excellent introduction about previous work of the group on the correlation between species in living in harsh conditions and possible specific genes that somehow could confers resistance to heat, cold, drought and salt stress.
Molecular biology methods and structural and functional Analysis of CYP2E1 Sequence are correct, well described and using appropriate websites and links.
Scientific names such as Lepus yarkandensis but also Saccharomyces cerevisiae, Atriplex canescens, Betula platyphylla and others should be written always in italica throughout the entire manuscript.
Figure 2 contains abundant information, but I recommend adding some of this information as supplementary material to simplify the presentation and allow a detailed observation to readers interested in those details. A detailed description of the main non conservative amino acid changes would be desirable.
Figure 4 needs a statistical treatment to identify the differences between the response of control cells and those expressing CYP2E1. Paragraph lines 228.244. is a qualitative description, but some parameters show a very similar pattern in both sets, suggesting that CYP2e1 expression does not matter (i.e. SOD or MDA). Protein damage seems to be higher in the presence of CYP2E1 in al mannitol concentrations. This pattern is unclear unless a statistical analysis will prove the hypothesis. Same comment concerning Figure 5 and 6.
For instance, considering Figure 6 (temperature tolerance), CAT activity tends to decrease as temperature is increased. MDA tends to increase. Concerning other biochemical markers related to oxidative stress, there are no difference between control cells and those expressing the CYP2E1 enzyme. Thus, it is difficult to assign a role to this enzyme in the resistance to high temperature.
In summary, the study has two different parts.
a) Molecular features of the gene and the encoding enzyme in comparison to CYPE2E1 from other species.
b) Effect of the heterologous expression in yeasts on the tolerance of cells to harsh conditions.
The second part is much more important, as is the part related to the title, but the conclusion is not clearly demonstrated. Statistical analysis is essential.
Figure 7 shows a promising scheme, but the pattern of some parameters is not congruent with the hypothesis. According to SOD and CAT increase, ROS should be decreased. This divergence is not justified.
Concerning discussion, some statements are not congruent with the abstract results. Two examples:
Line 359: “The results show that CYP2E1 protein has one transmembrane helix, which is consistent with its characteristics as a membrane--bound enzyme”. This opposite to the statement at the abstract.
Line 390 “The study also found that the expression of the CYP2E1 gene significantly increased gene significantly increased the activity of SOD and CAT in yeast cells”. This is not demonstrated in results.
Under my view, discussion about plants is a little bit long.
In case of an eventual substantial revision of the manuscript, think about this question Is the enzyme in that species more efficient than in other species in spite of the similarities? Some of the species mentioned at Figure 2 (including humans) cannot survive at the arid conditions of this hare. It is clear that CYP2E1 may be involved in yeast metabolism and detoxification processes, thereby protecting yeast cells to some extent, but the mechanism for that function remains unclear, and the protection to ROS and oxidative damage is not fully demonstrated.
Minor point: Methods for stress Resistance Testing of Transgenic Yeast Cells.
Line 528: “0.5, 1, 2, and 3 mol/L NaCl solutions were prepared with 0.01M PBS solution to simulate alkaline conditions”. NaCl creates salty rather than alkaline conditions. NaCl does not increase pH. “Alkaline” should be replaced by “salty” to be precise. This point should be considered at conclusion (see line 552, saline-alkali) as both terms are not synonyms. Consider this point throughout the manuscript.
Reviewer 2 Report
Comments and Suggestions for Authors
The study is significant in exploring the stress-adaptive role of the CYP2E1 gene in Lepus yarkandensis. However, clearer articulation of its broader applications, especially in biodiversity conservation and biotechnology, is needed.
Some sections, particularly in the Introduction and Discussion, contain excessive details that obscure the main message. A more concise presentation would improve readability.
The introduction effectively highlights the importance of the CYP2E1 gene, but a clearer connection to the study’s objectives is needed.
Clearly label and caption all figures with details experimental procedure. For instance, Figure 6 could include explanations for trends observed under temperature stress.
In the section on oxidative stress markers, discuss why Pro and MDA levels showed opposing trends in some cases.
Expand on the implications of CYP2E1 expression in other organisms or its potential applications in environmental biotechnology in discussion.
Round 2
Reviewer 1 Report
Comments and Suggestions for Authors
Both the reply letter and the modifications introduced in the manuscript are valuable and satisfy most of my concerns about the manuscript.
I still have some points to clarify the consistency between the replies contained in the cover letter and the modifications introduced at the manuscript. Some of the replies are not addressed in the revised version.
1) About comment 3. Authors reply that they have incorporated statistical analyses to clarify the differences between control and CYP2E1-expressing cells (Figures 4, 5, and 6). According to the reply letter, these analysis strenghenr the conclusions regarding the role of CYP2E1 in enhancing yeast cell tolerance to various stress conditions (lines 235-262; 271-299, 307-354 at the amended manuscript). However, ther is no description of the statistical analysis used at the method section, and there is no mention to p values or any other statistical parameter throughout the manuscript. Is there some misunderstanding about this point?.
2) About the discusion on the amino acid changes by conservative (homologous) residues (i.e. V/A at position 85) or not conservative residues (i.e C/S at position 268). The discusion is poor, not enough to be satisfactory. The last change might produce important modifications on the structural conformation of the protein as disulfide bridges could be modified. To 13 13 modifications in 293 amino acid is poor.
2) About the document on supp. figures. Figures are not visible at the doc document, and the format of the pdf document is not correct. Please, check it.
Thank you.
Reviewer 2 Report
Comments and Suggestions for Authors
No further issue.
Author Response
- First and foremost, I would like to extend my sincere gratitude for the time and effort you have dedicated to reviewing our manuscript titled "Functional Identification of CYP2E1 Gene in the Kidney of Lepus yarkandensis." We are pleased to hear that you find our research design appropriate, methods well-described, results clearly presented, and conclusions supported by the data.
- We have carefully considered your review and are confident that our study meets the journal's standards in terms of English language quality, background provided in the introduction, research design, methodological description, and the support of conclusions by results. As you have not raised any further issues or suggestions, we believe our manuscript is ready to proceed to the next stage.
- Once again, thank you for your valuable time and professional feedback. We look forward to contributing to the academic community with our research and anticipate further collaboration with MDPI.
Round 3
Reviewer 1 Report
Comments and Suggestions for Authors
It is OK. Thanks